# Sensory Integration Disorders in Patients with Multiple Sclerosis

**DOI:** 10.3390/jcm11175183

**Published:** 2022-09-01

**Authors:** Krystian Mross, Marta Jankowska, Agnieszka Meller, Karolina Machowska-Sempruch, Przemysław Nowacki, Marta Masztalewicz, Wioletta Pawlukowska

**Affiliations:** Department of Neurology, Pomeranian Medical University, 71252 Szczecin, Poland

**Keywords:** sensory integration, sensory integration disorder, multiple sclerosis, sensory processing, somatosensory sensitivity

## Abstract

Sensory integration disorder (SID) is also called, interchangeably, sensory processing disorder (SPD). Multiple sclerosis (MS) is an autoimmune, chronic, neurological disease of the central nervous system. Sensorimotor function disorders are present in both multiple sclerosis and SID. The study aimed to assess the SID among patients with MS and included 141 patients with relapse-remitting MS and 72 participants in the control group. To assess SID in both groups, a questionnaire prepared by Daniel Travis was used. Additionally, participants answered questions regarding their age, gender, handedness and in the study group about the duration of the disease, relapses in the past year and the advancement of the disease using EDSS. The occurrence of sensory seeking was significantly more frequent in the MS patients with relapses in the past year. Patients with MS had more often general disorders of sensory integration in the past. However, healthy subjects significantly more often showed the severity of social and emotional disorders in the past. Currently, the group of MS patients has a greater intensity of sensor-based motor abilities. The study revealed more severe SID in MS patients than in the control group. Still, more research is needed in this field.

## 1. Introduction

Sensory integration disorder (SID), which is called, interchangeably, sensory processing disorder (SPD) [1], was proposed by Anna Jean Ayres in 1972 [2] and was an attempt to define dysfunction in children, in which sensory stimuli were responded with atypical behavioural reactions. According to her work, adaptive behaviour lies upon adequate sensory integration (or processing), which is an unconscious process that is concerned with organizing information by one’s gustatory, visual, tactile, olfactory, proprioceptive and vestibular senses, and evaluating every acquired stimulus to decide on which to focus on and find an appropriate behavioural response to it [2,3]. Although understanding of neuroscience in her time was limited, current research continues to support the findings of her work [4].

There is still little research regarding the neuroanatomical underpinnings of SID. A key role is attributed to basal ganglia, cerebellum, neocortex and neural circuitries between them [1,5]. A functional foundation may however be attributed to deviation in the interplay of the vestibular and somatosensory system, which in response can further disrupt proper praxis and sensory modulation. This results in subjects exhibiting exaggerated responses to sensation in a hyper- or hyporesponsive manner, as well as sensory seeking behaviours, poor sensory discrimination and poor sensory-based motor abilities [4,6]. Eventually, they will be reflected as attention, movement and/or balance deficits [4]. Previously mentioned hyperresponsivity may be, according to a recent study, the result of lower GABA levels in the thalamus and the subsequent effect on atypical thalamic connectivity [7].

One of the most well-established approaches to SPD has been proposed by Winnie Dunn and her four quadrant sensory processing model [8]. According to this model, neurological threshold and behavioural response are the dimensions from which sensory processing is composed. High neurological threshold refers to individuals that are hyposensitive and may represent two patterns of behaviour, depending on their response strategies: (1) sensation seeking—individual represents active response strategy and will seek stimulus-rich environments in order to enhance the response; (2) low registration—the individual represents a passive response strategy and shows either slow or no response at all to stimuli. With regard to hypersensitivity and the associated low neurological threshold, two more patterns can be distinguished: (3) sensation avoiding—active response strategy that results in avoiding sensations, which are not comfortable for the concerned individual; (4) sensation sensitivity—passive response strategy to sensation that might be uncomfortable [6,8]. Accurate sensory processing provides a smooth response to the demands of the environment, while a disturbed one will interfere with the daily functioning of the affected individual in areas such as activities of daily living, instrumental activities of daily living, rest and sleep, education, work, play, leisure, and social participation [9].

According to the latest research, men may be more prone to having poor sensory integration. Moran et. all conducted a study to investigate sex differences and sensory integration function between male and female collegiate athletes using the modified Clinical Test for Sensory Interaction and Balance (m-CTSIB). Results of the study revealed that female athletes performed significantly better than male athletes on baseline m-CTSIB composite scores (*p* < 0.001) [10]. When it comes to handedness, Chinese researchers from the School of Psychology of Shanghai University of Sport reported that the inhibitory effect of the left hemisphere was significantly stronger compared to the right hemisphere. The study found that the dorsolateral prefrontal cortex of the left hemisphere is more prominent in inhibiting the ipsilateral primary motor cortex [11]. It is well known that handedness is determined by the dominance of one hemisphere, and this led us to consider how sensory integration might be impacted by right- or left-handedness. Those two studies allowed us to have a suspicion that gender and handedness are also independent factors that affect sensory integration.

Multiple sclerosis (MS) is an autoimmune, chronic, neurological disease of the central nervous system (CNS), leading to demyelination, neuronal damage, axonal loss and ultimately to corresponding neurological, cognitive and/or psychological deficits [12,13,14]. The underlying cause of multiple sclerosis is an inflammatory process whose pathogenesis is still not fully understood. The most established theory recognizes CNS antigen-specific CD4+ T cells in the periphery as the main causative factor. However, neither the exact localization of activation of those lymphocytes nor specific antigens responsible for this activation are known. It is also hypothesized that possible cross-reactivity, on the basis of molecular mimicry, with antigens of foreign nature, e.g., during viral infection, might also result in autoreactivity of CD4+ T cells [15]. Thus far, it is recognized that autoreactive T cells are capable of migrating through the blood–brain barrier, and while being present in the CNS, they might become reactivated by locally persisting antigen-presenting cells. This triggers the creation of an inflammatory cascade, which leads to the recruitment of additional inflammatory-related cells and the constant activation of macrophages. This ultimately results in axonal injury, as a consequence of oligodendrocyte loss and myelin damage [13].

Secondary involvement in the course of MS immunopathology is linked to CD8+ T cells, which are probably responsible for relapses in the course of MS. Another important role is attributed to B cells and antibodies. The serum of about 30% of MS patients might contain antibodies of demyelinating potential. Among them, those against potassium channel KIR 4.1, neurofascin and contactin-2 are now considered to be the most specific for MS. The role of B cells has been neglected until recently, and thus far, they are known for being responsible, i.a., for upholding proinflammatory differentiation of T cells [15].

The traditional classification of multiple sclerosis distinguishes four clinical phenotypes: relapsing remitting (RRMS), secondary progressive (SPMS), primary progressive (PPMS) and progressive relapsing (PRMS). However, clinical practice shows that this theoretical classification, based mainly on a patient’s symptoms and description of events from the past, is sometimes unable to adequately categorize the patient’s phenotype, as all multiple sclerosis phenotypes share common characteristics. Unfortunately, it has not yet been possible to create a single clinical, imaging, or laboratory characteristic that could clearly distinguish between the different types of the disease. There are some relative differences in a number of imaging and laboratory markers (CSF and serum neurofilament levels, rate of new lesion formation, rate of brain and spinal cord atrophy), but none of them allow us to precisely indicate the subtype of multiple sclerosis [14].

Natural history studies have presented that the vast majority of patients are diagnosed with a relapsing–remitting type of multiple sclerosis [14]. This form is mainly characterized by fatigue, difficulty balancing or walking, numbness, pain and difficulty remembering. Patients might also report difficulty concentrating, vision problems, urinary incontinence or urgency, depression, stiffness, dizziness, constipation, mood swings, irritability, sexual dysfunction, difficulty with speech and many others [16].

When it comes to the sensory disturbances that a patient may experience in the course of MS, the most commonly observed are central or musculoskeletal pain, tingling and numbness in their extremities, back and head. Furthermore, allodynia, which is a result of the mechanical or cold stimulus, and paraesthesia were also noted. Also worth mentioning are dysaesthesia, Lhermitte’s sign, trigeminal neuralgia and decreased vibration sensation [17].

Further considering symptomatology, postural and balance deficits are exponents of pathology in the vestibular system [18]. These symptoms are present in both multiple sclerosis [19] and sensory integration disorder [4,6], as well as sensorimotor function disorders, expressed by exteroceptive sensory impairment and widely understood motor dysfunction [6,20,21]. To date, little research has been carried out on sensory integration disorders in multiple sclerosis, although there has been a noticeable increase in interest in recent years. Studies to date have been performed mainly using specialized diagnostic equipment and were mainly aimed at showing correlations between neurophysiological exponents of sensory integration and the disease multiple sclerosis. Citing them all would be well beyond the scope of this article. Still, it has been shown, among other things, that for example, reduced somatosensory gating responses were correlated with mobility impairment [22]. Furthermore, slowing of the spinal somatosensory conduction, measured with evoked response potentials, results in reduced walking velocity [23]. Both of these aforementioned studies indicate difficulties in sensory processing. Similar conclusions about the link between motor disorders and poor sensory integration seem to come from studies conducted by Conte et al. and Gianni et al. [24,25]. Both of the studies showed lower index finger abduction velocity during somatosensory temporal discrimination threshold testing compared to healthy subjects. An interesting approach to stimulus processing impairment was presented in a study by Giurgiola et al., in which participants performed a simultaneity judgment task to assess the temporal binding window (TBW). Participants in the study group were significantly more likely to show enlarged audio–visual TBW than the control group, indicating asynchronous visual and auditory stimuli integration [26]. Considering cognitive abilities in MS and their impact on sensory integration, a study by researchers from Seville may be of interest. A visual task was performed by the participants during which a time–frequency analysis of electroencephalography (EEG) had been performed simultaneously. Reaction time and accuracy were worse in MS patients in comparison to healthy controls and delay in latency and lower amplitude in MS patients in evoked and induced alpha compared to the control group were observed in EEG, which suggested deficits in early sensorial and cognitive processing [27].

However, most of the previously mentioned studies did not use an established sensory processing model. In addition, those studies were more of a laboratory than clinical character. The authors of this manuscript, as of the date of this writing, are aware of only three studies in which the Dunn model and related questionnaires were used to assess MS patients. Colbeck’s study was directed at examining how sensory processing and cognitive fatigue affect the quality of life of MS patients. The patterns of low registration, sensation avoidance and sensory sensitivity were shown to predispose to high cognitive fatigue and low quality of life, while sensation seeking was shown to predispose to low cognitive fatigue and high quality of life [28]. Engel-Yeger et al. took as their goal to compare two groups of MS patients, one with cognitive impairment and the other without, with a control group in terms of functional capacity. The results showed that MS patients exhibit SPD characteristics regardless of their cognitive abilities and that exhibiting impairment in sensory processing translates into greater disease severity [29]. Furthermore, Stern et al. investigated the relationship between sensory processing patterns, trait anxiety and health-related quality of life. Their study found that the sensory sensitivity, sensation avoiding, and low registration patterns showed greater trait anxiety compared to the general population, and were also shown to have a lower mental and physical health-related quality of life. In contrast, the sensation seeker pattern showed the opposite characteristics, suggesting a potential protective factor in MS. In addition, sensory processing patterns were proven to have effects directly and indirectly on physical health-related quality of life, as well as indirectly on mental health-related quality of life [30].

With all of the above in mind and given the scarcity of reports to date on the impact of sensory integration on the course of MS, our study aimed to assess the impact of sensory integration disorders on the course and manifestation of the disease in MS patients, to analyse the relationship between gender and handedness and sensory integration disorders in people with MS, and to compare the course of sensory integration disorders in healthy people and those affected by MS.

## 2. Materials and Methods

### 2.1. Material

The study involved 141 people diagnosed with relapse–remitting type MS according to the McDonald 2010/2017 criteria (94 women and 47 men; aged 25–45 years, mean age ± SD: 39.0 ± 10.41) who are patients of the drug treatment program for MS at the Department of Neurology of Independent Public Clinical Hospital No. 1 of the Pomeranian Medical University named Prof. Tadeusz Sokołowski in Szczecin. The line of treatment and the type of MS medication used by the patients were not analysed in this research.

The exclusion criteria concerning our study group were as follows:Primary progressive and secondary progressive forms of MS—this criterion was used to exclude the potential impact of the course of different types of MS on the results;SI disorders diagnosed in childhood;Mental illness;People before the age of 25 and over 45—we decided to include in the study only subjects not older than 45, because available studies show that sensory integration is changing over a lifetime [31].

### 2.2. Methods

The Daniel Travis questionnaire for the study of SI disorders in people over 18 years of age based on the guidelines of the American Society for SI Disorders in Adults was used to investigate SI disorders (Appendix A). It distinguishes the domains of General Modulation (9 statements), Over-Responsiveness (26 statements), Under-Responsiveness/Sensory Seeking (20 statements), Sensory Discrimination (26 statements), Sensory-Based Motor Abilities (19 statements) and Social and Emotional (22 statements). Each symptom was rated by the study participant using a 5-point Likert scale from 0 (never occurred) to 4 (regularly occurs). If the problem had occurred in the past, but no longer persists, the participant entered the letter P. The sum obtained by adding the values in each domain indicated the severity of the SI disorder in that domain.

In addition, the authors’ questionnaire was used to collect information on age, gender, duration of the disease, the severity of the disease assessed using the EDSS scale, handedness and the occurrence of relapses in the past year. EDSS was assessed during scheduled follow-up visits at our Department of Neurology by a neurologist specialising in the diagnosis and treatment of MS.

A relapse in the course of MS was diagnosed when a patient was presenting new neurological symptom(s) that could be correlated with new lesion(s) in the central nervous system, as detected on magnetic resonance imaging. An increase in EDSS scale value by at least 0.5 pts was also mandatory to reach such a diagnosis and was assessed by a neurological specialist who focuses on MS treatment.

### 2.3. Control Group

The control group consisted of healthy subjects (53 females and 19 males; aged 25 to 45 years, mean age ± SD: 36.0 ± 5.7) who were employees of Independent Public Clinical Hospital No. 1 of the Pomeranian Medical University named Prof. Tadeusz Sokołowski in Szczecin and who consented to participate in the study.

### 2.4. Statistical Analysis

Arithmetic means, minimum and maximum values, standard deviations, medians, lower and upper quartiles were used to describe numerical variables. To present qualitative data, the percentage and abundance of the variable were given. Levene’s test showed heterogeneity of variance in the study groups and patient subgroups (*p* < 0.05). For this reason, and because the distribution of variables deviated from a normal distribution (Shapiro–Wilk test, *p* < 0.05), the Mann–Whitney U test was used to compare groups of variables. Statistical significance was set at *p* < 0.05. The analysis was performed using Statistica version 13.3 (StatSoft Inc., Kraków, Poland), under a current licence.

## 3. Results

The characteristics of the patients included in the study and participants in the control group are presented in Table 1.

In the last year, a relapse occurred in 81.5% of MS patients. In this group, the occurrence of sensory discrimination (*p* = 0.0) was found to be significantly more frequent compared to the group that did not have a relapse in the last year (Table 2).

The differences in severity of sensory integration disorders between MS patients concerning the stage of the disease using the EDSS scale are presented in Table 3.

Next, the severity of sensory integration disorders was compared in patients with MS and those without the disease (control group) (Table 4).

Table 4 compares the group of people with MS with the group of healthy people. Both groups did not differ statistically in terms of sex and handedness. The group of MS patients was significantly older (*p* = 0.0). It has been shown that patients with MS more often in the past, i.e., before the onset of disease symptoms, had general disorders of sensory integration compared to the group of healthy people (*p* = 0.0 *). Moreover, healthy subjects significantly more often showed the severity of social and emotional disorders in the past compared to patients with MS (*p* = 0.0). Currently, the group of MS patients has a greater intensity of sensor-based motor abilities impairment compared to the group of healthy people (*p* = 0.0 *), which is taking into account the nature of the disease.

Additionally, a comparison of the degree of exposure to sensory integration disorders in right-handed people with MS (n = 125) and right-handed participants in the control group (n = 61) revealed statistically important differences. Healthy right-handed participants more often had impaired general modulation in the past compared to the right-handed patients with MS (points; mean ± SD; 0.3 ± 1.3 vs. 0.0 ± 0.2, *p* = 0.0) and social and emotional disorders in the past (points; mean ± SD; 0.4 ± 1.5 vs. 0.0 ± 0.2, *p* = 0.0). However, in comparison to the control group, the group of right-handed patients with MS more often showed the severity of sensory-based motor abilities (points; mean ± SD; 9.6 ± 8.8 vs. 16.1 ± 16.4, *p* = 0.0).

## 4. Discussion

Sensory integration disorders in MS patients mainly concern the motor and sensory spheres [32]. It seems crucial to investigate sensorimotor functions and their impact on the course of MS. Sensory and multisensory representations are seen as the ‘building blocks’ on which higher cognitive abilities and learning are built. The adult somatosensory cortex retains the ability to functionally modify neurons [33,34,35]. Body cortical maps can be modified by eliminating sensory stimuli or by behavioural training involving stimulation of skin mechanoreceptors. Brain or spinal cord injury in MS can manifest as sensory abnormalities that manifest as both hyper- and hypoaesthesia [36]. Topographical maps in the somatosensory cortex of adult primates are capable of radical reorganisation following peripheral nerve damage [37]. The somatosensory system is altered by demographic as well as by environmental factors that lead to functional and structural changes in the central nervous system [38,39,40]. Such changes may be due to neuronal noise due to increased basal impulse activity of nerves, which results in attenuation of the neuronal signal.

In our study group of patients with MS, a reduction in sensory discrimination has been found to occur in patients who have had a relapse in the past year. It may be hypothesised that patients with decreased sensory discrimination have a greater susceptibility to the onset of relapse [41]. This hypothesis may also be supported by the fact that the level of sensory discrimination changes in temperature between 30–60 °C [42]. In patients with MS, increased disease activity occurs during spring and summer. This fact is associated with temperature changes [43,44,45]. Disease relapses were less frequent in patients with less severity of sensory discrimination. This hypothesis requires further, more detailed research. Could discrimination disorders be a trigger for relapse or are they only a coincidental statistical relationship that is not confirmed by clinical images? One should also wonder whether the onset of relapse triggers a decrease in sensory discrimination. Relapses are focal lesions in the central nervous system that manifest themselves with a specific clinical presentation characteristic to the focus of the relapse. Patients with a relapse of MS do not complain of a sudden, general disturbance of sensory discrimination. This may suggest that sensory discrimination disorders do not appear after a relapse, and the lack of insight into their appearance in the patient may argue for a long-term adaptation of individuals to these changes. In such a case, taking into account all the aspects discussed, can we conclude that in people with MS who have a decreased threshold for sensory discrimination, external factors such as higher temperature activate the disease process?

Our hypothesis on the influence of integration disorders on the course of MS can also be confirmed by the fact that MS patients significantly more often had general sensory integration disorders in the past compared to the control group. Currently, general sensory integration disorders are not different in MS patients compared to controls. Can we, therefore, conclude that the sensory integration disorders appeared even before the onset of the symptoms of the disease? Could this indicate that sensory integration disorders precede motor symptoms, and could they influence the course of MS?

Popa (2013) et al. [46] reported that dysfunctions in the cerebellum can lead to disruptions in the coherence between the sensory and motor cortex. Furthermore, the cerebellum plays a huge key role in sensorimotor coupling [47]. Hence, perhaps it is the disruption in sensorimotor processing that leads to the later changes that occur in MS? We believe that our preliminary study should be the start to further, larger studies confirming or rejecting this thesis.

We also showed a difference between the current and past sensorimotor functioning of people with MS and in the control group in terms of social and emotional contacts. Patients with MS significantly less frequently reported impaired social and emotional contacts in the past compared to the control group. It is known that patients with MS have limited social activity [48]. Our study group showed no such changes. Could this indicate a high adaptation to the disorders and a lack of criticism regarding the judgement of impairments in social interactions and emotional experiences? Are MS patients at all able to self-assess disturbances in emotional–social processing? The test that was used in the study is a test of subjective sensations assessed currently and in the past. Integration of different modalities occurs early in information processing to ensure a multisensory and holistic experience of affective information [49]. Multisensory integration accelerates and improves our ability to understand the emotions of others, and thus, multimodal presentation of emotional information may support emotion perception in MS. Although impaired emotional processing is an inherent feature of MS, our study group reported much fewer emotional–social disorders in the past than in the control group [48,49]. This lack of criticism may confirm early damage to the structures responsible for socio-emotional processing. Available studies have reported the emergence of emotional disturbances in the early stages of the disease [50]. Our study suggests that socio-emotional disturbances may be another marker predisposing to the disease. As shown in animal studies, sensory discrimination disorders coincide with socio-emotional disorders [51]. In the group with MS, we have differences in the processing of social–emotional stimuli in the past as well as sensory discrimination in the past compared to the control group. It is during the early stages of development that transient neurons and circuits form the scaffolding for the development of neural networks [52]. Temporary changes in activity during crucial periods of development can lead to permanent changes in functional connectivity and may therefore underlie the manifestation of neurological and psychiatric conditions [53]. Endogenous activity in the somatosensory system often results from central pattern generators in motor areas. Early activity in any sensory system can influence activity in areas of the sensory cortex, whereas deprivation can lead to altered intermodal connections. Early circuits dominated by transient neural cell types coexist with circuits that will dominate in adults for some time. Many neurodevelopmental disorders, such as autism spectrum disorders and schizophrenia, are associated with an imbalance of excitatory and inhibitory neurons in the prefrontal cortex [54,55]. Therefore, could early disorders of somatosensory integration impart an implication for the onset of MS? Studies have shown that spontaneous and sensory-evoked activity directly regulates axonal myelination [56,57]. Furthermore, altered neuronal activity may also contribute to inflammation in the immature human brain [58]. Could a similar process occur in adults? Our study represents a preliminary hypothesis that should be supported by objective studies.

### Study Limitations

The study is a subjective judgement of somatosensory parameters in patients with MS.The evaluation should be long-term and carried out on an even larger number of patients.

## 5. Conclusions

In preliminary studies, it can be concluded that impaired sensory integration affects the occurrence and course of MS.Patients with MS will start to have general sensory processing disorders more often and less often emotional and social disorders in the past compared to the group of healthy people.Patients with MS who had experienced a relapse in the past year showed greater sensory discrimination disorders compared to patients without relapses.

## Figures and Tables

**Table 1 jcm-11-05183-t001:** Characteristics of the patients with MS in the study group and healthy participants in the control group.

Parameters		All Patients with MS (n = 141)	Control Group (n = 72)
**Demographic data**			
Age (years; mean ± SD)		40.2 ± 10.4	36.3 ± 5.7
Sex	females	94 (66.6%)	53 (73.6%)
	males	47 (33.3%)	19 (26.3%)
**Characteristics of the groups**			
Handedness	right-handedness	125 (88.6%)	61 (84.7%)
	left-handedness	16 (11.3%)	11 (15.2%)
EDSS Scale (points; mean ± SD)		1.9 ± 1.2	_
Duration of the disease (years; mean ± SD)		0.8 ± 0.8	_
Number of relapses in the last year (mean ± SD)		1.1 ± 0.6	_
Relapses in the past year	yes	115 (82.1%)	_
	no	25 (17.8%)	

MS, multiple sclerosis; EDSS, Expanded Disability Status Scale; SD, standard deviation.

**Table 2 jcm-11-05183-t002:** A detailed analysis of the severity of sensory integration disorders in patients with MS with relapses during the last year and without relapses.

Variables	Patients with MS with Relapses in the Last Year (n = 115)	Patients with MS without Relapses in the Last Year (n = 25)	*p*-Value
General modulation (points; mean ± SD)	9.7 ± 7.1	8.3 ± 6.9	0.3
General modulation in the past (points; mean ± SD)	0.0 ± 0.2	0.1 ± 0.4	0.1
Over-Responsiveness(points; mean ± SD)	23.6 ± 17.9	22.4 ± 20.5	0.5
Over-Responsiveness in the past (points; mean ± SD)	0.1 ± 0.4	0.4 ± 0.7	0.1
Under-Responsiveness/Sensory Seeking (points; mean ± SD)	18.9 ± 12.2	15.4 ± 11.0	0.1
Under-Responsiveness in the past /Sensory Seeking in the past (points; mean ± SD)	0.2 ± 0.5	0.2 ± 0.7	0.3
Sensory Discrimination(points; mean ± SD)	17.5 ± 17.2	10.0 ± 12.5	**0.0 ***
Sensory Discrimination in the past (points; mean ± SD)	0.2 ± 0.9	0.0 ± 0.0	0.3
Sensory-Based Motor Abilities (points; mean ± SD)	16.7 ± 16.2	12.3 ± 14.0	0.1
Sensory-Based Motor Abilities in the past (points; mean ± SD)	0.1 ± 0.4	0.0 ± 0.2	0.6
Social and Emotional (points; mean ± SD)	21.5 ± 17.8	17.6 ± 14.9	0.3
Social and Emotional in the past (points; mean ± SD)	0.0 ± 0.3	0.0 ± 0.2	0.9

MS, multiple sclerosis; SD, standard deviation; * *p* < 0.05.

**Table 3 jcm-11-05183-t003:** The differences in the severity of sensory integration disorders between MS patients concerning the stage of the disease using the EDSS scale.

Variables	Group of Patients with MS up to 2.5 Points on the EDSS scale (n = 118)	Group of Patients with MS 3 or more Points on the EDSS scale (n = 23)	*p*-Value
General modulation (points; mean ± SD)	9.4 ± 7.2	10.0 ± 5.9	0.4
General modulation in the past (points; mean ± SD)	0.0 ± 0.3	0.0 ± 0.2	0.9
Over-Responsiveness (points; mean ± SD)	22.3 ± 18.0	29.1 ± 18.8	0.1
Over-Responsiveness in the past (points; mean ± SD)	0.2 ± 0.5	0.1 ± 0.6	0.4
Under-Responsiveness/Sensory Seeking (points; mean ± SD)	18.8 ± 12.3	15.7 ± 10.1	0.4
Under-Responsiveness in the past /Sensory Seeking in the past (points; mean ± SD)	0.2 ± 0.5	0.2 ± 0.5	0.9
Sensory Discrimination (points; mean ± SD)	15.3 ± 16.5	21.0 ± 16.8	0.1
Sensory Discrimination in the past (points; mean ± SD)	0.2 ± 0.9	0.0 ± 0.2	0.9
Sensory-Based Motor Abilities (points; mean ± SD)	14.0 ± 14.6	26.9 ± 18.3	**0.0 ***
Sensory-Based Motor Abilities in the past (points; mean ± SD)	0.1 ± 0.4	0.0 ± 0.0	0.4
Social and Emotional (points; mean ± SD)	20.5 ± 16.7	23.1 ± 20.5	0.7
Social and Emotional in the past (points; mean ± SD)	0.0 ± 0.3	0.0 ± 0.0	0.7

MS, multiple sclerosis; EDSS, Expanded Disability Status Scale; SD, standard deviation; * *p* < 0.05.

**Table 4 jcm-11-05183-t004:** The differences in the degree of intensification of sensory integration disorders between patients with MS and healthy subjects (control group).

Variables		Group of MS Patients (n = 141)	Control Group (n = 72)	*p*-Value
Age (years; mean ± SD)		40.2 ± 10.4	36.3 ± 5.7	0.0 *
Sex	females	94 (66.6%)	53 (73.6%)	0.2
	males	47 (33.3%)	19 (26.3%)	
Handedness	right-handedness	125 (88.6%)	61 (84.7%)	0.4
	left-handedness	16 (11.3%)	11 (15.2%)	
General modulation (points; mean ± SD)		9.5 ± 7.0	10.0 ± 5.9	0.3
General modulation in the past (points; mean ± SD)		0.0 ± 0.3	0.3 ± 1.2	**0.0 ***
Over-Responsiveness (points; mean ± SD)		23.4 ± 18.3	22.8 ± 15.1	0.8
Over-Responsiveness in the past (points; mean ± SD)		0.1 ± 0.5	0.8 ± 3.3	0.4
Under-Responsiveness/Sensory Seeking (points; mean ± SD)		18.3 ± 12.0	20.9 ± 12.5	0.1
Under-Responsiveness in the past /Sensory Seeking in the past (points; mean ± SD)		0.2 ± 0.5	0.4 ± 1.1	0.1
Sensory Discrimination (points; mean ± SD)		16.2 ± 16.6	15.7 ± 14.4	0.6
Sensory Discrimination in the past (points; mean ± SD)		0.1 ± 0.8	0.0 ± 0.4	0.1
Sensory-Based Motor Abilities (points; mean ± SD)		16.1 ± 16.0	10.7 ± 10.2	**0.0 ***
Sensory-Based Motor Abilities in the past (points; mean ± SD)		0.1 ± 0.3	0.0 ± 0.3	0.3
Social and Emotional (points; mean ± SD)		20.9 ± 17.3	23.8 ± 15.6	0.1
Social and Emotional in the past (points; mean ± SD)		0.0 ± 0.3	0.3 ± 1.4	**0.0 ***

MS, multiple sclerosis; SD, standard deviation; * *p* < 0.05.

## Data Availability

The data cannot be made publicly available due to privacy regulations.

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
