# Peer review of "Sensory Integration Disorders in Patients with Multiple Sclerosis"

_jcm, 2022, doi:10.3390/jcm11175183_

Round 1

Reviewer 1 Report

This is an interesting manuscript to read. SID is pretty new for me and I learned from this manuscript (and extra readings). I believe that this study may be interesting to some clinicians.

However, there are a number of issues that I think can be improved to make the manuscript easier to read.

1/ Daniel Travis questionnaire should be added in the supplement to help those who are not familiar with this and to understand the contents

2/ How do people with MS identify their EDSS as most of them would not have any ideas about EDSS, which requires a specialist's knowledge to identify

3/ Why PPMS is excluded? how about SPMS? why is the age limit at 45? 

4/ Lines 86-90: I was not sure the main ideas of this long sentence.

5/ Lines 96-97: I do not agree that motor disorders in MS or slow walking speed are the results of SID. Ref #20 (Arpin et al.) suggest that "altered somatosensory gating responses were correlated with the mobility impairments" (in PwMS).

6/ Line 164: a typo?

7/ line 244-245: was there any evidence that MS disease activity increases during spring and summer? 

8/ Lines 246-247: "Relapses did not occur in patients with MS who did not have sensory discrimination disorders". Is this true? MS people who did not experience relapse still had SID but less frequent as suggested in Table 2

9/ Line 249: this is perhaps a bold hypothesis.

10/ lines 289-290: "Our study suggests that socio-emotional disturbances may be another marker predisposing to the disease". Does this contradict the results shown in Table 4, where socioemotional points in the past are lower in people with MS compared with healthy control?

Reviewer 2 Report

In this paper, the authors evaluated the sensory integration disorders in patients with Multiple Sclerosis.

The work has many deficiencies for its publication.

The introduction explains little about the pathology of multiple sclerosis, it does not expose the different types, different symptoms, different evolution according to the patient and different sensory alterations.

Materials and Methods: has the treatment of MS patients been taken into account? First or second line drugs? clarify.

How has the type of outbreaks of each patient been considered? Clarify

Discussion: first introductory paragraph again about sensory integration, without talking about EM at all.

The type of flare in relapses in MS patients is not specified.

The authors pose questions to which they do not give a clear answer.

Round 2

Reviewer 1 Report

Thank you for clarifying the points in question in the revised manuscript. The hypotheses raised from this pilot research may be worth further research to confirm (or reject) the hypotheses. 

Reviewer 2 Report

The authors have resolved all issues raised in the review.